# Optimization Model and Application for Agricultural Machinery Systems Based on Timeliness Losses of Multiple Operations

**Jian Sun** [1,2,†], **Yiming Zhang** [1,2,†], **Haitao Chen** [1,2,3,*] **and Jinyou Qiao** [1,2,*]

1    College of Engineering, Northeast Agricultural University, Harbin 150030, China; 18231538638@163.com (Y.Z.)
2    Heilongjiang Province Technology Innovation Center of Mechanization and Materialization of Major Crops Production, Harbin 150030, China
3    College of Mechanical and Electronic Engineering, East University of Heilongjiang, Harbin 150066, China
*    Correspondence: htchen@neau.edu.cn (H.C.); jyqiao@neau.edu.cn (J.Q.)
†    These authors contributed equally to this work.

**Abstract:** Present agricultural practices confront issues such as mismatches between tractors and implements, imprecise machinery allocation, and excessive machinery investment. Optimization of agricultural machinery systems was a potent remedy for these concerns. To address inaccuracies in calculating objective functions and the incompleteness of constraints in existing models for agricultural machinery system optimization, a comprehensive mixed integer nonlinear programming (MINP) model for agricultural machinery system optimization was established. The model introduced timeliness loss costs for multiple key operations across various crops into the objective function, and constraints were enhanced by including operation sequence constraints and boundary constraints on initiation and completion dates of those key operations. Taking corn and soybeans as examples, timeliness loss functions of sowing and harvesting operations were derived through experiments. Solving the MINP model by Lingo (V.14.0) software, improvements in total power, workload per unit power, and total operational costs were shown when comparing the optimized machinery system through the MINP model against current systems. When the model omitted considerations for timeliness loss functions and operation sequence constraints, issues arose including an increase in total operational costs and an inversion of operation sequence. The model's application in devising machinery allocation plans for production units of various operational scales revealed a gradual decrease in total power and costs per unit area with expanding scale, approaching stability when scale exceeded 1600 hm$^2$. This study enriches theory and methodology for optimizing agricultural machinery systems, provides theoretical and technological underpinnings for rational machinery acquisition, and promotes the high-quality progression of comprehensive agricultural mechanization.

**Keywords:** agricultural machinery system; mixed integer nonlinear programming (MINP); timeliness loss; operation sequence; operating scale





## 1. Introduction

The optimization of agricultural machinery systems is an important basis for realizing the high-quality development of the whole process of agricultural mechanization, and also an important technical support for the transformation and upgrading of agricultural mechanization to the direction of quality and efficiency [1]. The development of a scientifically robust model for optimizing agricultural machinery systems, and effectively leveraging these systems to boost the efficiency and economic benefits of mechanized agricultural production, is essential for the realization of modern agriculture and meets the urgent demand for high-quality agricultural mechanization development.

Research in the field of agricultural machinery system optimization started early. In the 1960s, American Professor Hunter began applying operational research theories to optimize agricultural machinery systems [2]. By the 1970s, based on a different operating

scale and diverse crop production processes, the types and quantities of agricultural machinery were selected with a goal to minimize costs. However, problems such as slowed computational speed and inadequate accuracy remained [3]. With the widespread application of operations research methodologies and computer technology, significant advancements had been introduced in agricultural machinery system optimization model construction, as well as in speed and precision of model solving. Typically, the lowest cost was used as the objective function, and the quantity of agricultural machinery and operation volumes were adopted as the constraints. Linear programming was a commonly used modeling method in the field of agricultural machinery system optimization. This kind of method had been applied in both developing countries like China, Brazil, and Iran [4–7], and developed countries such as the United States, the United Kingdom, Spain, Poland, and Italy [8–12].

Owing to the differences in crop varieties, natural environments, and other factors, each crop had its unique growth and maturity cycle. This resulted in strict seasonality in various agricultural operation stages, especially for various key operations. Performing key operations at inappropriate times, which could lead to a reduction in crop yields and quality degradation, are known as timeliness losses. Due to the timeliness loss rate functions of crops that could only be obtained through experiments, it was not easy to apply the function to solve the optimization of agricultural machinery systems. The timeliness loss function was generally a nonlinear function. Therefore, when considering the timeliness losses of key crop operations in the agricultural machinery system optimization model, the model would be a nonlinear programming model. Most reports studied concentrate exclusively on the timeliness losses of singular operations like sowing or harvesting. To date, researchers such as Meng Fanqi et al. in China hypothesized about the relationship between wheat sowing time delay and yield loss [13]. Zhou Yingchao et al. in China applied the timeliness loss rate coefficients for wheat and corn sowing to replace the timeliness loss function. While this approach achieved the goal of solving the nonlinear programming model for optimizing agricultural machinery systems, the optimization results still contained inaccuracies [14]. Other researchers, such as Sorensen in Denmark, took into account the timeliness losses during the harvesting stage of wheat and sugar beet. The number of harvesters were allocated when building the agricultural machinery system optimization model for the harvest stage [15]. Toro et al. in Sweden utilized two decades of farm yield data to analyze variations in wheat harvest timeliness losses under different weather conditions and established a nonlinear agricultural machinery system optimization model. Through simulation, the optimal harvest time and quantity of agricultural machinery were determined [16,17]. Omrani A et al. in Iran suggested that the number of days of mechanized operations for sugarcane harvesting is one of the essential factors for calculating and determining the model and quantity of machinery and their timeliness costs [18]. Wang Jinwu, Wang Guimin et al. in China conducted an experiment on the timeliness loss of rice harvesting operations and explored the relationship between the timeliness loss quantity of rice and the number of combine harvesters. The insights for the rational allocation of combine harvesters during the rice harvest period was offered through their findings [19–21]. Some scholars had conducted research focusing on machinery allocation for various operations in production units. For example, Gao Huanwen et al. in China considered the impact of the timeliness loss for wheat harvest and corn sowing on operation costs. Due to a lack of relevant data, they only proposed to reduce crop losses from delayed operations by shortening the operation time as much as possible [22]. Khani et al. in Iran proposed the cost function of the timeliness loss for corn harvest and wheat sowing according to the ASABE standard. And a mathematical model to determine the optimal operation time of key operations was established providing a reference for the construction of nonlinear programming optimization models for agricultural machinery systems [23]. Vatsa et al. in India established an agricultural machinery selection model considering the timeliness loss of rice, wheat, and corn sowing and harvesting. The types and quantities of agricultural machinery were analyzed, which should be equipped under

different scale conditions [24]. Rafael C.T. et al. in Brazil established a cost-energy demand agricultural machinery system optimization model for soybean and corn rotation areas and analyzed and compared various planting areas as examples, verifying the reliability of the optimization model [25]. Qiao Jinyou et al. in China obtained the law of timeliness loss for soybean sowing [26] and harvesting [27,28] through experiments, and introduced them into the objective function of the nonlinear programming model. The model was applied to the soybean and corn rotation area in Heilongjiang Province, and the optimization results were significant [29].

The application of linear and nonlinear programming methods to optimize agricultural machinery systems were widespread, and considerable achievements have been approached in theoretical methods and model solving. Nonlinear programming model that considered the timeliness loss of crops was more conformed to the requirements of agricultural machinery system optimization. However, existing research did not fully consider the constraints, such as constraints on the operation sequence and boundary dates of key operations. There were few reports on obtaining the regulations of timeliness loss of multiple key crop operations and applying regulations to the optimization model of agricultural machinery systems. Owing to the machinery units not needing to work all the time for a whole day to ensure that the operation was completed with the minimum number of machinery, which made the part of variables not integers. However, the final result of the number of tractors and implements must be integers. Therefore, the optimization model most met the practical requirements when it was a mixed integer programming model. To sum up, the constraints of the model were refined, and a model agricultural machinery systems based on Mixed Integer Nonlinear Programming (MINP) considering the timeliness loss of multiple operations was constructed. The timeliness loss of multiple key crop operations were derived through experiments to solve the model. The results provided a practical machinery system allocation scheme for the research area, and enriched and improved the theory and methodology in this field.

## 2. Materials and Methods

### 2.1. Four-Dimensional Subscript Variable Setting

Given the agronomic requirements for various operations, there was often an overlap in the timing of each operation, resulting in a greater number of agricultural stages compared to operation tasks. According to principles established in the literature [30], when constructing an optimization model for agricultural machinery systems, variables should be set separately for each agricultural stage. As the same operation might be performed by machinery units composed of different models of tractors and implements, and the same machinery unit could participate in different operations for various crops during different agricultural stages, therefore, a four-dimensional subscript setting should be adopted for the machinery unit variables in the agricultural machinery system optimization model. The variable $X_{ijkl}$ represented the quantity of machinery units composed of the $j$th type of tractor and the $k$th type of operation machinery, performing operations on the $l$th type of crop during the $i$th agricultural stage. Here, $i$ represented the sequence number of the agricultural stage, with subscript values ranging from 01 to 99; $j$ represented the sequence number of the tractor, with subscript values ranging from 00 to 99—the value of $j$ is 00 when it was a self-propelled machinery such as a combine harvester; $k$ represented the sequence number of the implement, with subscript values ranging from 11 to 99; $l$ represented the sequence number of crop type, with subscript values ranging from 01 to 99.

### 2.2. Objective Function Establishment

The optimization model of an agricultural machinery system typically aimed to minimize the operation cost. The objective function of the MINP model included three components: the annual fixed cost of machinery, the annual variable cost of each machinery unit, and the annual timeliness loss cost of the crop's key operations. Therefore, the objective

function of the MINP agricultural machinery system optimization model was shown in Equation (1).

$$\min C = C_f + C_v + C_{tl} \tag{1}$$

In the equation, $C_f$ represented the annual fixed cost of machinery (CNY 10,000); $C_v$ represented the annual variable cost of the machinery unit (CNY 10,000); and $C_{tl}$ represented the annual timeliness loss cost of key crop operations (CNY 10,000).

### 2.2.1. Model of Annual Fixed Cost of Machinery

The annual fixed cost of machinery comprised the machinery's annual depreciation cost and the annual management fee. Considering the large amount of machinery investment and the long lifespan of machinery, the time value of the invested capital should be taken into account. Hence, the dynamic depreciation method was adopted to calculate each machine's annual depreciation cost [31]. However, due to differences in the depreciation period and salvage value rate of tractors and implements, the annual fixed cost models for tractors and implements needed to be established separately. The models for tractors and implements are shown in Equations (2) and (3), respectively.

$$C_{ft} = \sum_{j=j_1}^{j_m} \left\{ \left[ \left[ P_{tj}(1+I)^{L_{tj}} - P_{tj}S_{rt} \right] \times \frac{I}{(1+I)^{L_{tj}} - 1} + \alpha P_{tj} \right] X_j \right\} \tag{2}$$

$$C_{fm} = \sum_{k=k_1}^{k_m} \left\{ \left[ \left[ P_{mk}(1+I)^{L_{mk}} - P_{mk}S_{rm} \right] \times \frac{I}{(1+I)^{L_{mk}} - 1} + \alpha P_{mk} \right] X_k \right\} \tag{3}$$

In these equations, $C_{ft}$ represented the annual fixed cost of the tractor (in CNY 10,000); $j_1$ represented the 1st type of tractor; $j_m$ represented the type of tractor within the machinery systems; $P_{tj}$ represented the purchasing price of the $j$th type of tractor (CNY 10,000); $I$ represented the discount rate (%); $L_{tj}$ represented the depreciation period of the $j$th type of tractor (years); $R_{st}$ represented the salvage value rate of the tractor (%); $\alpha$ represented the ratio of the annual management cost of machinery to the purchasing price of the machinery (%); $X_j$ represented the quantity of the $j$th type of tractor required to complete operations throughout the year; $C_{fm}$ represented the annual fixed cost of the implement (in CNY 10,000); $k_1$ represented the 1st type of implement; $k_m$ represented the type of implement within the machinery systems; $P_{mk}$ represented the purchasing price of the $k$th implement (in CNY 10,000); $L_{mk}$ represented the depreciation period of the $k$th implement (years); $R_{sm}$ represented the salvage value rate of the implement (%); and $X_k$ represented the quantity of the $k$th implement required to complete operations throughout the year.

### 2.2.2. Model for Annual Variable Cost of Operation Machinery Units

The variable cost for the operation machinery units were the sum of variable costs across all machinery units within the agricultural machinery systems, which were determined by the operation time, quantity of machinery units, and daily variable cost of the machinery units. As the variable cost varies for the same machinery unit when conducting different operations, and the types and quantities of machinery units performing the same operation may differ across agricultural stages, the variable cost should be represented distinctly for different operations and agricultural stages. The variable cost model for the operation machinery units is shown in Equation (4).

$$C_v = \sum_{j=j_1}^{j_m} \sum_{k=k_1}^{k_m} C_{vjk} = \sum_{i=q_s}^{q_e} \sum_{j=j_1}^{j_m} \sum_{k=k_1}^{k_m} \sum_{l=l_1}^{l_m} D_i X_{ijkl} C_{ijkl} A_{ijkl} \tag{4}$$

In this equation, $C_{vjk}$ represented the annual variable cost (in CNY 10,000) of the machinery unit composed of the $j$th type of tractor and the $k$th type of operation machinery; $q_s$ represented the starting agricultural stage of the $q$th operation task; $q_e$ represented the ending agricultural stage of the $q$th operation task; $l_1$ represented the 1st type of

crop; $l_m$ represented the type of crop within the production unit; $D_i$ represented the duration of the $i$th agricultural stage (days); $C_{ijkl}$ represented the variable cost per unit area (CNY 10,000/hm$^2$) of the machinery unit, consisting of the $j$th type of tractor and the $k$th type of operation machinery, performing operations for the $l$th crop during the $i$th agricultural stage; $A_{ijkl}$ represented the operating efficiency (hm$^2$/day) of the machinery unit, comprising the $j$th type of tractor and the $k$th type of operation machinery, for the $l$th crop during the $i$th agricultural stage.

### 2.2.3. Model for Timeliness Loss Cost of Key Operations

The timeliness loss cost of crops related to crop type, timeliness loss function for key operations, optimal crop yield, operation area for each agricultural stage, and crop sale price. The calculation model for the timeliness loss cost of key operations is shown in Equation (5). In this equation, the timeliness loss function for key operations should be determined through appropriate experimental trials, selecting suitable crop varieties within the model application region. When crop varieties and the planting region were determined, the operation time of key crop operations would affect the growth and development of crops and then affect the timeliness loss of crops. When sowing in advance, insufficient soil temperature would reduce the emergence rate of crops. Delaying sowing or harvesting in advance would lead to an insufficient crop growth cycle, resulting in an insufficient crop maturity. Delaying harvesting would also consume crops' own nutrients after maturity, reducing the quality and yield of crops.

$$C_{tl} = \sum_{l=l_1}^{l_m} P_l \sum_{p=l_{p_1}}^{l_{pm}} \sum_{i=q_{lps}}^{q_{lpe}} \left( \sum_{j=j_1}^{j_m} \sum_{k=k_1}^{k_m} \left( Y_{lp\max} X_{ijkl} A_{ijkl} \int_{T_{ilps}}^{T_{ilpe}} y_{lp}(t)dt \right) \right) \tag{5}$$

In this equation, $P_l$ represented the sale price of the $l$th agricultural product (CNY/kg); $l_{p_1}$ represented the 1st key operation of the $l$th crop; $l_{pm}$ represented the $m$th key operation of the $l$th crop; $q_{lps}$ represented the starting agricultural stage of the $p$th key operation of the $l$th crop; $q_{lpe}$ represented the ending agricultural stage of the $p$th key operation of the $l$th crop; $Y_{lp\max}$ represented the yield on the optimal operation date of the $p$th key operation of the $l$th crop (kg); $T_{ilps}$ represented the start time of the $p$th key operation for the $l$th crop during the $i$th agricultural stage; $T_{ilpe}$ represented the end time of the $p$th key operation for the $l$th crop during the $i$th agricultural stage; and $y_{lp}(t)$ represented the timeliness loss rate function for the $p$th key operation of the $l$th crop.

### 2.3. Constraints of MINP Optimization Model

### 2.3.1. Operation Area Constraint

The operation area constraint ensured that the sum of the operation quantities of all machinery units across each agricultural stage was equal to or exceeds the total operation quantities for that task. The operation area constraint model is shown in Equation (6).

$$\forall_q \left( \sum_{i=q_s}^{q_e} \sum_{j=j_1}^{j_m} \sum_{k=k_1}^{k_m} \sum_{l=l_1}^{l_m} \left( D_i A_{ijkl} X_{ijkl} \right) - S_q \right) \geq 0 \tag{6}$$

In this equation, $S_q$ represented the total operation quantity for the $q$th task (hm$^2$).

### 2.3.2. Tractor Allocation Constraint

The tractor allocation constraint indicated that the quantity of any given type of tractor should correspond to the maximum number of this model allocated across all agricultural stages. The tractor allocation constraint model is shown in Equation (7).

$$X_j - \forall_i \left\{ \sum_{k=k_1}^{k_m} \sum_{l=l_1}^{l_m} X_{ijkl} \right\} \geq 0 \tag{7}$$

### 2.3.3. Implement Allocation Constraint

The operation machinery allocation constraint indicated that the allocated quantity of any model of operation machinery should be equivalent to the maximum number of this model assigned across all agricultural stages. The implement allocation constraint model is shown in Equation (8).

$$X_k - \forall_i \left\{ \sum_{j=j_1}^{j_m} \sum_{l=l_1}^{l_m} X_{ijkl} \right\} \geq 0 \tag{8}$$

### 2.3.4. Operation Sequence Constraint

If there was an overlap in the operational time in any two operations, and the subsequent operation could only be conducted upon completion of the preceding operation, then the daily operation area of the subsequent operation should not exceed the sum of operation areas completed by the preceding operation before that day. This constraint guarantees that every operation could be executed as per agricultural requirements. The operation sequence constraint model is shown in Equation (9).

$$\begin{cases} \forall_{d_{it}} \left[ \sum_{i=q_s}^{q_e} \sum_{j=1}^{m_j} \sum_{k=11}^{m_k} \sum_{l=1}^{m_l} \left( D_{q_s} A_{q_s jkl} X_{q_s jkl} + (d_{(q_s+1)t} - 1) A_{(q_s+1)jkl} X_{(q_s+1)jkl} \right) - \sum_{i=(q+1)_s}^{(q+1)_e} \sum_{j=1}^{m_j} \sum_{k=11}^{m_k} \sum_{l=1}^{m_l} \left( d_{(q+1)_s t} A_{(q+1)_s jkl} X_{(q+1)_s jkl} \right) \right] \geq 0 \\ 0 \leq d_{it} \leq D_i, Math.cell(d_{it}) \\ d_{(q_s+1)t} - 1 = d_{(q+1)_s t} \end{cases} \tag{9}$$

In this equation, $d_{it}$ represented the $t$th day of the $i$th agricultural stage; $D_{q_s}$ represented the number of days for the $q$th operation during the initial agricultural phase; $A_{q_s jkl}$ represented the operational efficiency of the machinery unit, which was composed of the $j$th tractor and the $k$th operational machinery for the $l$th crop during the starting agricultural stage (hm$^2$/day) of the $q$th operation; $X_{q_s jkl}$ represented the quantity of machinery units that consist of the $j$th tractor and the $k$th implement, tasked with the $q$th operation for the $k$th crop during the initial agricultural stage; $d_{(q_s+1)t}$ represented the $t$th day for the 2nd agricultural stage of the $q$ operation; $A_{(q_s+1)jkl}$ represented the operational efficiency of the machinery unit, which was comprised of the $j$th tractor and the $k$th implement for the $l$th crop during the start of the 2nd agricultural stage of the operation (hm$^2$/day); $X_{(q_s+1)jkl}$ represented the quantity of machinery units comprised of the $j$th tractor and the $k$th implement, which carried out the operation for the $l$th crop at the beginning of the 2nd agricultural stage of the operation; $d_{(q+1)_s t}$ represented the $t$th day of the $q+1$th operation during the initial agricultural stage; $A_{(q+1)_s jkl}$ represented the operational efficiency of the machinery unit, which was comprised of the $j$th tractor and the $k$th implement during the beginning of the agricultural stage of the $q+1$th operation (hm$^2$/day); $X_{(q+1)_s jkl}$ represented the quantity of machinery units, which was comprised of the $j$th tractor and the $k$th implement, and carried out the operation for the $l$th crop during the start of the agricultural stage of the $q+1$th operation.

### 2.3.5. Boundary Constraint for Start and End Dates of Key Operations

There existed a regulatory relationship between the start and end dates of key crop operations and the quantity of machinery units required for these key operations. Therefore, even though the start and end dates of key operations were variables prior to optimization, they must still fall within the appropriate range. According to the optimal period distribution theorem [32], the upper and lower bounds of operation time could be determined based on the actual operation dates of key crop operations locally. The boundary constraint model for the start and end dates of key operations is shown in Equation (10).

$$\begin{cases} T_{lp\min} \leq T_{lps} \leq T_{lpopt} \\ T_{lpopt} \leq T_{lpe} \leq T_{lp\max} \end{cases} \tag{10}$$

In this equation, $T_{lp\min}$ represented the upper limit of the start date for the $p$th key operation of the $l$th crop; $T_{lps}$ represented the actual start date of the $p$th key operation for

the *l*th crop; $T_{lpopt}$ represented the optimal operation date for the *p*th key operation of the *l*th crop; $T_{lpe}$ represented the end date of the *p*th key operation for the *l*th crop; and $T_{lpmax}$ represented the upper limit of the end date for the *p*th key operation of the *l*th crop.

### 2.3.6. Non-Negative Variable and Integer Constraints

In the agricultural machinery systems, the number of each model of tractor, the quantity of implements, and the quantity of operation machinery units at each agricultural stage could not be negative. Therefore, all variables in the previous objective function and constraint equations were non-negative. The quantity of machinery units within the same agricultural stage could be non-integer because the operation task could be accomplished without requiring a full day's work. In actual production, the number of tractors and implements should be integers, which was the main reason that the model is a mixed-integer programming model. The non-negative variable constraint and integer constraint model are shown in Equation (11).

$$\begin{cases} X_j \geq 0, Math.cell\left(X_j\right) \\ X_k \geq 0, Math.cell\left(X_k\right) \\ \quad X_{ijkl} \geq 0 \end{cases} \tag{11}$$

## 3. Results

A 2000 $hm^2$ agricultural production unit in Harbin, Heilongjiang Province, was selected for study. This unit employs a "corn–corn–soybean" rotation system, with soybean planting covering an area of 666.7 $hm^2$ and corn covering 1333.3 $hm^2$.

### 3.1. Experiment on Timeliness Loss for Key Operations of Corn and Soybean

The functions of timeliness loss for key crop operations were an essential condition for solving the MINP optimization model of agricultural machinery systems, and these functions needed to experimentally determined. Considering the sowing and harvesting of corn and soybean as critical operational procedures in their life cycle, we followed the method in references [26,27] to conduct experiments on timeliness loss of sowing and harvesting operations. The experiments took place at the Xiangyang Base of Northeast Agricultural University (125°42′ E, 44°04′ N) in Xiangfang District, Harbin. The resultant timeliness loss functions provided a crucial basis for establishing a nonlinear optimization model for agricultural machinery systems.

### 3.1.1. Experimental Materials

We chose XY335 and HN55, which were suitable for planting in the Harbin area, as test varieties for corn and soybeans, respectively. High-concentration potassium sulfate compound fertilizer was applied simultaneously with sowing.

The PTQ-A3 electronic weighing scale (with a weighing accuracy of 0.1 g) from HuaZhi, a U.S. company, was selected to measure the weight of corn and soybean grains. Corn and soybean grains were dried using a DHG-9030A electric blast drying oven (with a temperature accuracy of ±0.1 °C) from Shanghai Yiheng Scientific Instrument Co., Ltd. (Shanghai, China). The grain moisture content was calculated thereafter.

### 3.1.2. Experimental Design

Based on the regular sowing and harvesting dates of the test varieties XY335 and HN55 in Harbin, the sowing and harvesting periods for the experiments were extended on both ends. The sowing operations' timeliness loss experiments for the two crops took place from 22 April to 16 May 2022, while the harvesting operations' timeliness loss experiments took place from 23 September to 17 October 2022. Each sowing or harvesting date was spaced 3 days apart. Both sowing and harvesting operations' timeliness loss experiments had nine operational period treatments, with each treatment being repeated three times for the four experiments. Each experiment involved 27 test plots, each plot being 5 m long

and 2.6 m wide (4 ridges). To remove the boundary effect, each test plot was spaced 1 m apart longitudinally, with a 5 m buffer at both ends of the experimental area and a 1.3 m (2 ridges) buffer on both sides. The total length of each experimental area was 63 m and the total width was 10.4 m, totaling an area of 655.2 m$^2$.

All plots for the timeliness loss experiments of corn and soybean harvesting operations were sown on 4 May. Based on the sowing operation times for each experiment and following the requirements for seed sowing depth and hole spacing, manual ridge sowing was adopted. All plots for the timeliness loss experiments of corn and soybean sowing operations were harvested on 5 October, with other cultivation measures and operation methods being consistent. This ensured that the sowing and harvesting dates were the sole variables for the timeliness loss experiments of the sowing and harvesting operations.

### 3.1.3. Test Method

Manual sampling was conducted according to the harvesting operation times mentioned above. For each plot in the timeliness loss experiments of corn and soybean, 5 m$^2$ and 2 m$^2$ areas were respectively selected to test the yield. After manual harvesting, corn cobs or soybean pods with stalks were promptly placed into net bags labeled with corresponding numbers. During manual threshing, we ensured that the grains were intact and completely threshed, and after threshing, the grains were placed into small net bags with corresponding numbers.

In the lab, an electronic weighing scale was adopted to measure the weight of corn or soybean grains. Corn and soybean grains were dried using an electric constant temperature blast drying oven. The actual moisture content of corn and soybean grains for different sowing dates was calculated, and the yield of corn and soybean in the test plots under standard moisture content was calculated according to the requirements of GB/T10362-2008 [33] and GB 1352-2023 [34].

The timeliness loss rate on a certain day was the percentage of the timeliness loss amount on that operation date out of the yield during the optimal operation period. The timeliness loss amount on a specific operation date was the difference between the yield during the optimal operation period and the yield on that operation date. The timeliness loss rate for each operation date is as follows.

$$LR_{lpt}(\%) = \frac{Y_{lp\max} - Y_{lpt}}{Y_{lp\max}} \times 100 = \frac{Y_{lp\max} - \frac{10^4}{S_{lp}} \cdot \frac{M_{lpt}\left(1 - F_{lpt}\right)}{1 - F_{lb}}}{Y_{lp\max}} \times 100 \qquad (12)$$

In this equation, $LR_{lpt}$ represented the timeliness loss rate (%) of the $p$th key operation of the $l$th crop during the $t$th test; $Y_{lp}$ represented the yield of crop grains under the standard moisture content in the $t$th test area of the $p$th key operation of the $l$th crop (kg/hm$^2$); $S_{lp}$ represented the test harvest area (m$^2$) of the $p$th key operation of the $l$th crop; $M_{lp}$ represented the actual grain weight (kg/m$^2$) of the test area of the $p$th key operation of the $l$th crop; $F_{lpt}$ represented the actual grain moisture content (%) of the $t$th test of the $p$th key operation of the $l$th crop; and $F_{lb}$ represented the standard moisture content of the $l$th crop grains.

### 3.1.4. Determination of Timeliness Loss Functions of Key Operations

IBM SPSS Statistics 23 software was utilized to perform a nonlinear regression analysis on the timeliness loss rate of crops and operation date, confirming that the timeliness loss rates of the four operations—corn and soybean sowing and harvesting—change in a quadratic function over time. Figure 1 presents the timeliness loss rate functions for these operations for the corn and soybean test varieties.

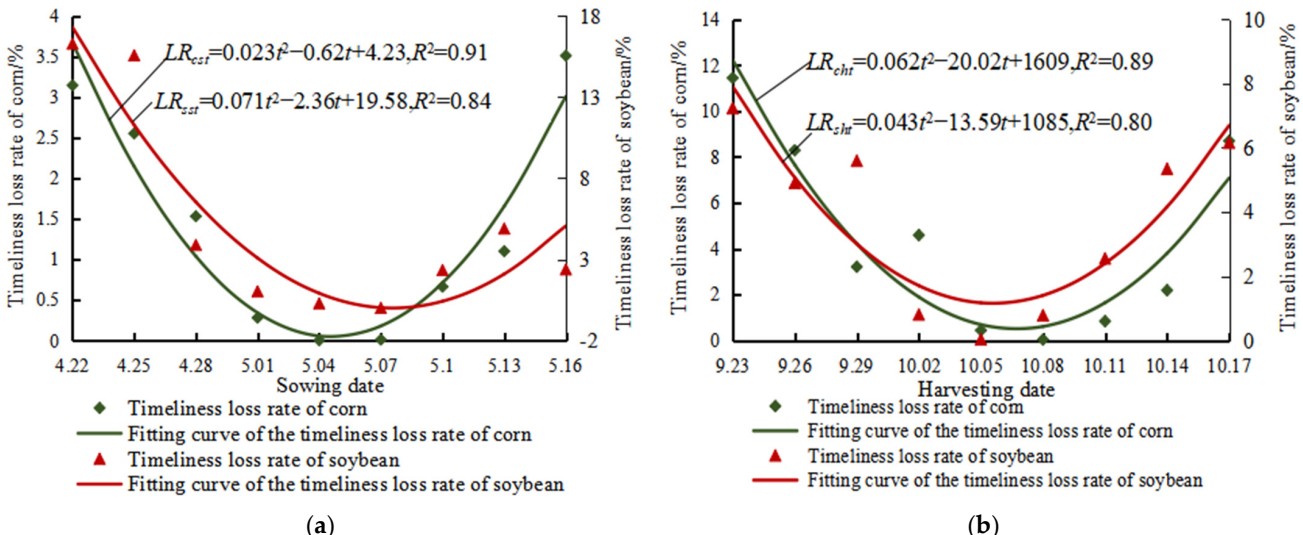

**Figure 1.** Timeliness loss rate functions of soybean and corn. (**a**) Sowing operations; (**b**) Harvesting operations.

From Figure 1, it could be inferred that the optimal sowing dates for XY335 corn and HN84 soybean in Harbin were 4 May and 7 May, respectively. The optimal harvesting dates for XY335 corn and HN84 soybean in Harbin were 7 October and 4 October, respectively. Early or delayed sowing and harvesting operations would result in a loss of crop timeliness. The longer the interval between the operation time and the optimal operation date, the higher the timeliness loss rate.

*3.2. Corn–Soybean Rotation and Rotational Tillage Production Process*

The survey found that the production unit employs a rotational tillage strategy for mechanized agriculture, involving two years of no-tillage and sowing with straw mulching followed by a year of combined tillage. The approach for handling corn stubble involves the use of no-tillage and sowing with straw mulching technology. The 2BFMJ series of no-till straw mulching precision seeders were employed for the technology, capable of executing seven operations in a single pass, namely, "clearing and preventing straw blockage, preparing the seedbed, applying deep side fertilization, seeding in place, covering and rolling the soil, spraying pesticides, and ensuring even straw coverage" [35]. The technology not only resolved the difficulties of corn stubble treatment in the northeast region, but also completed multiple operations simultaneously to save time and operation costs. However, perennial no-tillage and sowing with straw mulching may lead to problems like soil crusting and stubble residues. Hence, after two consecutive years of no-tillage for corn stubble, the soybean stubble was handled using combined tillage in the third year. This method could mix crushed soybean straw into the 0–20 cm soil layer, simultaneously carrying out deep loosening of 30–40 cm. The process caused minimal soil disturbance to maintain moisture, and after deep loosening, the soil could facilitate water permeation underground during the flood season and used soil capillarity to bring underground water to the surface during dry seasons. Soil water storage capacity was improved, and the purpose of drought resistance and increased yield was achieved [36]. Simultaneously, the no-tillage straw mulching precision seeder could perform precision seeding without the stubble-clearing device. The full-mechanized operation process is shown in Figure 2.

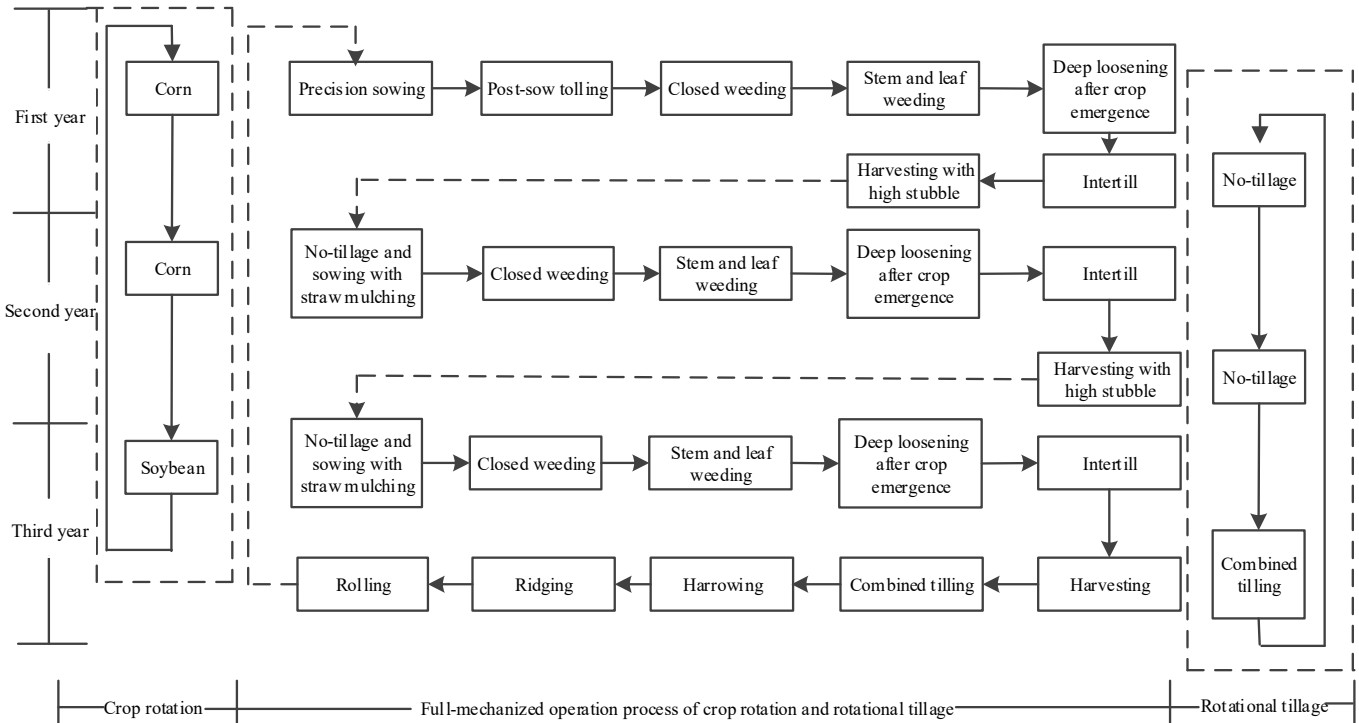

**Figure 2.** Full-mechanized operation process of crop rotation and rotational tillage.

### 3.3. Determination of Agricultural Machinery Models and Parameters

The production unit currently owned four types of tractors, including two models produced in China and two produced in other countries, that is, $m_j = 4$. There were 3 types of combine harvesters, all produced in other countries, and 13 other types of operation machinery, all produced in China, that is, $m_k = 16$.

Upon investigation, parameters such as the depreciation years, salvage value rate, and purchase price of the agricultural machinery in the production unit were obtained. When the discount rate, $I = 4.9\%$, and management fee rate, $\alpha = 1\%$, were taken into account, the above parameters could be substituted into Equations (2) and (3) to calculate the fixed costs of tractors and implements, respectively. With the operation process of crop rotation and rotational tillage shown in Figure 2 and the actual survey of machinery unit operation situations, the daily production rates, $A_{ijkl}$, and variable costs of each operation machinery unit, $C_{v_{jk}}$, were determined. The agricultural machinery variable number, operation parameters, and operation costs are shown in Table 1.

**Table 1.** Agricultural machinery number, operation parameters, and operation costs.

| Tractors | | | | | Implements | | | | | | Operation Machinery Units | | |
|---|---|---|---|---|---|---|---|---|---|---|---|---|---|
| Variables | Power (kW) | $P_{tj}$ (CNY 10,000) | $L_{tj}$ (years) | $C_{ft}$ (CNY 10,000) | Variables | Kind | The Width of the Implements (m) | $P_{mk}$ (CNY 10,000) | $L_{mk}$ (years) | $C_{fm}$ (CNY 10,000) | Operating Item | $C_{ijkl}$ (CNY/hm²) | $A_{ijkl}$ (hm²/d) |
| $X_{01}$ | 66.15 | 12.37 | 13 | 1.39 | $X_{11}$ | No-till straw mulching precision seeders | 3.25 | 14 | 8 | 2.22 | No-tillage and sowing with straw mulching | 17.16 | 124 |
| | | | | | | | | | | | Precision sowing | 20.59 | 111.60 |
| | | | | | $X_{13}$ | Rollers | 8.5 | 3.74 | 8 | 0.59 | Rolling | 69.12 | 19.47 |
| | | | | | $X_{15}$ | Sprayers | 12 | 3.85 | 8 | 0.61 | Weeding | 61.09 | 17.33 |
| | | | | | $X_{17}$ | Culti-vators | 3.6 | 3.8 | 8 | 0.60 | Deep loosening, intertill, and ridging | 28.96 | 66.67 |
| $X_{02}$ | 95.6 | 18.13 | 13 | 2.03 | $X_{11}$ | No-till straw mulching precision seeders | 3.25 | 14 | 8 | 2.22 | No-tillage and sowing with straw mulching | 19.50 | 121.37 |
| | | | | | | | | | | | Precision sowing | 23.40 | 109.24 |
| | | | | | $X_{14}$ | Rollers | 12.8 | 5.6 | 8 | 0.89 | Rolling | 99.36 | 22.13 |
| | | | | | $X_{16}$ | Sprayers | 18 | 6.3 | 8 | 1 | Weeding | 84.00 | 20 |
| | | | | | $X_{18}$ | Culti-vators | 5.4 | 4.8 | 8 | 0.76 | Deep loosening, intertill, and ridging | 43.52 | 73 |
| $X_{03}$ | 154.35 | 68 | 16 | 6.72 | $X_{12}$ | No-till straw mulching precision seeders | 5.85 | 19.6 | 8 | 3.10 | No-tillage and sowing with straw mulching | 35.10 | 111.23 |
| | | | | | | | | | | | Precision sowing | 42.12 | 100.11 |
| | | | | | $X_{19}$ | Culti-vators | 7.7 | 7.2 | 8 | 1.44 | Deep loosening, intertill, and ridging | 68.96 | 84 |
| | | | | | $X_{23}$ | Combined tillage machines | 3.6 | 18.5 | 8 | 2.93 | Combined tilling | 30.40 | 216 |
| | | | | | $X_{24}$ | Heavy harrows | 6.2 | 6.8 | 8 | 1.08 | Harrowing | 55.52 | 84 |
| $X_{04}$ | 271 | 131 | 16 | 12.95 | $X_{25}$ | Combined tillage machines | 6.8 | 31 | 8 | 4.91 | Combined tilling | 43.20 | 300 |
| | | | | | $X_{26}$ | Heavy harrows | 7.8 | 7.1 | 8 | 1.12 | Harrowing | 81.92 | 100 |
| | | — | | | $X_{20}$ | Combine harvesters | 155 | 98 | 16 | 9.69 | Harvesting | 23.66 | 261.20 |
| | | — | | | $X_{21}$ | | 239 | 275 | 16 | 27.18 | | 31.55 | |
| | | — | | | $X_{22}$ | | 284 | 206 | 16 | 20.36 | | 50.08 | |

### 3.4. Model Optimization Results

By integrating the operation requirements of crop rotation and rotational tillage in Figure 2, the set variables and related parameters were inputted into Equations (1)–(11). After a specific formulation, the MINP agricultural machinery system optimization model included 224 variables. There were 524 constraint conditions from seven sets of constraint equations, including 12 nonlinear constraint equations.

Lingo is the abbreviation for linear interactive and general optimizer—interactive linear and general optimization solver. It is a comprehensive set of tools designed to quickly, conveniently, and effectively construct and solve linear, nonlinear, and integer optimization models. The MINP agricultural machinery system optimization model was solved using Lingo (V.14.0) [37]. The results showed that the total operation costs of the current machinery systems and the MINP model optimization were CNY 3.4861 million and CNY 2.8587 million, respectively. Compared to the current machinery systems, the total operation costs decreased by 18.00% by MINP model optimization. The number and fixed costs of tractors and combine harvesters in the current machinery systems and the MINP model optimization are shown in Table 2.

**Table 2.** Changes in machinery quantity and fixed costs.

| Machinery Type | Variables | Number of Machines | | Fixed Cost (In CNY 10,000) | |
|---|---|---|---|---|---|
| | | Current | MINP | Current | MINP |
| Tractors | $X_{01}$ | 3 | 3 | 4.17 | 4.17 |
| | $X_{02}$ | 3 | 1 ↓ | 6.09 | 2.03 ↓ |
| | $X_{03}$ | 5 | 4 ↓ | 33.60 | 26.88 ↓ |
| | $X_{04}$ | 1 | 0 ↓ | 12.95 | 0.00 ↓ |
| Subtotal | | 12 | 8 ↓ | 56.81 | 33.08 |
| Seeders | $X_{11}$ | 3 | 1 ↓ | 6.66 | 2.22 ↓ |
| | $X_{12}$ | 4 | 4 | 12.40 | 12.40 |
| Subtotal | | 7 | 5 ↓ | 19.06 | 14.62 ↓ |
| Rollers | $X_{13}$ | 2 | 2 | 1.18 | 1.18 |
| | $X_{14}$ | 1 | 0 ↓ | 0.89 | 0.00 ↓ |
| Subtotal | | 3 | 2 ↓ | 2.07 | 1.18 ↓ |
| Sprayers | $X_{15}$ | 1 | 1 | 0.61 | 0.61 |
| | $X_{16}$ | 2 | 1 ↓ | 2 | 1 ↓ |
| Subtotal | | 3 | 2 ↓ | 2.61 | 1.61 ↓ |
| Cultivators | $X_{17}$ | 5 | 3 ↓ | 3.00 | 1.80 ↓ |
| | $X_{18}$ | 3 | 1 ↓ | 2.28 | 0.76 ↓ |
| | $X_{19}$ | 3 | 3 | 3.42 | 3.42 |
| Subtotal | | 11 | 7 ↓ | 8.70 | 5.98 ↓ |
| Combine harvesters | $X_{20}$ | 2 | 0 ↓ | 19.38 | 0.00 ↓ |
| | $X_{21}$ | 2 | 0 ↓ | 54.36 | 0.00 ↓ |
| | $X_{22}$ | 2 | 3 ↑ | 40.72 | 61.08 ↑ |
| Subtotal | | 6 | 3 | 114.46 | 61.08 ↓ |
| Combined tillage machines | $X_{23}$ | 3 | 3 | 8.79 | 8.79 |
| | $X_{24}$ | 1 | 0 ↓ | 4.91 | 0.00 ↓ |
| Subtotal | | 4 | 3 ↓ | 13.70 | 8.79 ↓ |
| Heavy harrows | $X_{25}$ | 3 | 2 ↓ | 3.24 | 2.16 ↓ |
| | $X_{26}$ | 1 | 0 ↓ | 1.12 | 0.00 ↓ |
| Subtotal | | 4 | 2 ↓ | 4.36 | 2.16 ↓ |
| Total | | 50 | 32 ↓ | 221.77 | 128.50 ↓ |

Note: "↑" represented an increase in the number of machines or fixed costs compared to current machine systems; "↓" represented a reduction in the number of machines or fixed costs compared to current machine systems.

The data in Table 2 indicated that following optimization by the MINP model, the tractor of 271 kW power was not selected, the total amount of tractors reduced from 12 to 8, and annual fixed costs were saved by 41.77%. The optimization results suggested that agricultural production units with a farming scale of 2000 hm$^2$ should not opt for high-powered tractors of 271 kW or more. In terms of machinery selection, the optimization results of the model showed that the number and types of rollers, combine harvesters, combined tillage machines, and heavy harrows have all decreased. Notably, 12.8 m rollers of 6.2 m width, a 284 kW combine harvester, 6.8 m combined tillage machines of 6.2 m width, or 7.8 m heavy harrows of 6.2 m width were not selected in the optimized models. Specifically, there was a reduction of one roller, three combine harvesters, one combined tillage machine, and two heavy harrows, with the annual fixed costs decreasing by 43.00%, 46.64%, 35.84%, and 50.46%, respectively. This suggests that these four types of machinery were over-represented in the current machinery systems. Following optimization, the types of seeders, sprayers, and cultivators remained the same, but their total numbers were reduced by two, one, and four units, respectively, and their annual fixed costs decreased by 23.29%, 38.31%, and 31.26%, respectively. Overall, the MINP model optimization resulted in an overall reduction of 18 units of machinery, 6 types of machinery, and a decrease in fixed costs by 42.06%.

The total power of the machinery systems in agricultural machinery was the total power of tractors and combine harvesters, which in the current machinery systems and the MINP optimized scheme were calculated to be 2884.00 kW and 1763.45 kW, respectively. Compared with the current machinery systems, in optimization by the MINP model, the total power of the machinery systems decreased by 38.85%, and the power burden per unit area increased by 63.77%, significantly reducing energy consumption and improving the utilization rates of tractors and combine harvesters.

## 4. Discussion

### 4.1. Analysis of the Impact of Timeliness Loss Rate Function on Total Operation Cost

If the optimization model of the agricultural machinery systems did not take into account the timeliness loss rate function, the model would be a mixed integer linear programming (MILP) model. The total operation cost of the MILP agricultural machinery system optimization model was CNY 2,445,700, which was 14.45% lower than the total operation cost of the MINP model. However, the total operation cost of the MILP agricultural machinery system optimization model at this time consists of two parts: the annual fixed cost of the machinery and the variable cost of the machinery unit. Therefore, the total cost of the MILP agricultural machinery system optimization model needed to be recalculated after adding the timeliness loss cost. The number and operation time of machinery units at each agricultural stage could be determined by solving the model. The crop timeliness loss rate function in Figure 1, the operation efficiency parameters, number, and operation time of machinery units were substituted into Equation (5) where the timeliness loss cost could be calculated. The comparison of costs after optimization by the two models are shown in Figure 3.

As could be seen from Figure 3, the variable costs of the two models were similar, mainly because the operation area completed within the machinery systems was the same. Compared with the optimization by the MILP model, the variable costs of the MINP model increased by 8.75%, mainly because the variable cost per unit area of different machinery units completing the same operation was different. There were also differences in the types and quantities of agricultural machinery of the two models. Compared with the results of the MILP model, the fixed costs of the MINP model increased by 6.48%, and the timeliness loss costs decreased by 74.79%. This was mainly because the MILP model did not take into account the timeliness loss cost. The model only required that the number of machinery systems could complete the required operation area of all operations, which reduced the fixed costs of the machinery systems but leads to a prolongation of the critical operation time of corn and soybeans, and a substantial increase in the timeliness loss cost.

Considering the three operation costs, the total operation cost of the MILP model was 16.96% higher than that of the MINP model. Therefore, the calculation results of the MINP model were more in line with reality.

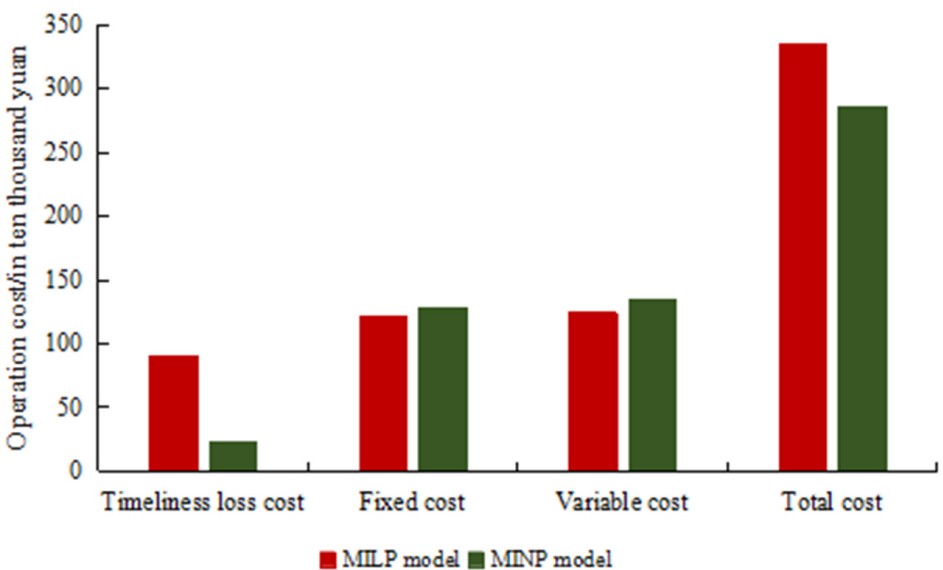

**Figure 3.** Comparison of operation cost composition.

*4.2. Analysis of the Impact of Operation Sequence Constraints on Optimization Results*

Most current research on the optimization model of the agricultural machinery systems did not consider operation sequence constraints, which could not ensure that the daily operation volume of each operation follows agronomic requirements. Lingo (V.14.0) software was applied to solve the MINP agricultural machinery system optimization model without operation sequence constraints. The results were compared with those of the MINP model with operation sequence constraints. The comparison found that the total operational costs and the quantity of each type of machinery were the same, but there were differences in the operation area completed at different agricultural stages for the same operation. The precise sowing of corn and post-sow rolling of corn were taken as examples. As these two operations overlap in time with other operations, they span across five agricultural stages. The start and end times of each agricultural stage can be derived from the optimization results, and the daily operational area and cumulative operational area of the two operation projects can be calculated under two conditions as the operation time changes. The results are shown in Table 3.

**Table 3.** Cumulative Operation Area of Each Operation.

| Agricultural Stage | $I_1$ | $I_2$ | | $I_3$ | | $I_4$ | | $I_5$ | |
|---|---|---|---|---|---|---|---|---|---|
| Working date (month·day) | | 5.2 | 5.3 | 5.4 | 5.5 | 5.6 | 5.7 | 5.89 | 5.9 | 5.10 |
| Daily sowing area (hm²) | 77 | 84 | 84 | 84 | 84 | 84 | 85 | 85 | |
| Cumulative sowing area (hm²) | 77 | 161 | 245 | 329 | 413 | 497 | 582 | 667 | |
| Daily post-sow tolling area without restraint conditions (hm²) | | 138 | 29 | 28 | 28 | 28 | 138 | 138 | 140 |
| Cumulative post-sow tolling area without restraint conditions (hm²) | | 138 | 167 | 195 | 223 | 251 | 389 | 527 | 667 |
| Daily post-sow tolling area under restraint conditions (hm²) | | 77 | 56 | 56 | 56 | 56 | 122 | 122 | 122 |
| Cumulative post-sow tolling area under restraint conditions (hm²) | | 77 | 133 | 189 | 245 | 301 | 423 | 545 | 667 |

Note: The operation time marked in the table refers to the completion date of the operation.

As could be seen from Table 3, the daily operation area within the same agricultural stage for the same operation project was the same, and there were cases where the daily operation area between different agricultural stages for the same operation project was the same. This result also proved that the variables in the optimization model needed to be set separately according to the agricultural stage. Post-sow tolling of corn needed to be completed the day after corn sowing, that is, the cumulative operation area of corn sowing on the same day should not be less than the cumulative operation area of post-sow tolling of corn on the following day. If operation sequence constraints were not included in the optimization model, 77 hm$^2$ of corn sowing would be completed on 2 May, which was less than the 138 hm$^2$ of post-sow tolling completed on 3 May; 161 hm$^2$ of corn sowing would be completed by 3 May, which was less than the 167 hm$^2$ of post-sow tolling completed by 4 May. Therefore, there were plots for post-sow tolling before complete sowing, and the order of sowing and post-sow rolling operations in these plots had been reversed. If operation sequence constraints were included in the optimization model, the daily operation area of the two operations could meet agronomic requirements. This verified the necessity of setting operation sequence constraints in the optimization model of the agricultural machinery systems.

### 4.3. Analysis of Machinery Allocation Variation Rules for Different Operation Scales

According to the above MINP agricultural machinery system optimization model, the crop rotation and rotational tillage mode of the production unit, and the related data of the machinery unit, the agricultural machinery allocation and operation cost changes for production scales of 400~4000 hm$^2$ were calculated at a gradient of 400 hm$^2$. The number of tractors and combine harvesters and the power per unit area for different production scales are shown in Figure 4.

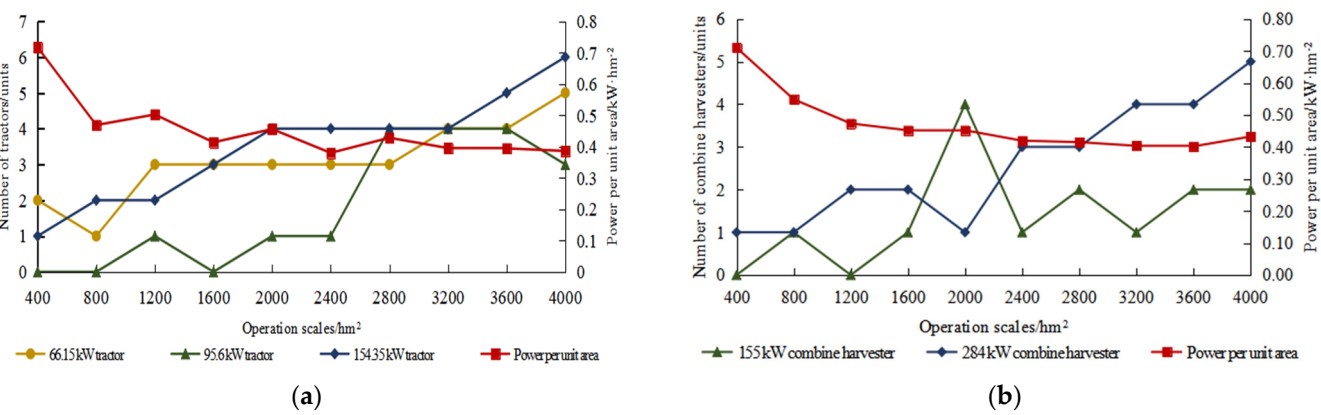

**Figure 4.** Number and power per unit area of tractors and combine harvesters. (**a**) Tractors; (**b**) Combine harvesters.

As could be seen from Figure 4, the tractors of 271 kW power were not chosen for different production scales after optimization by the MINP model. This was mainly because under the studied production mode, the tractors of 271 kW power participated in fewer operation projects, resulting in poorer economics. Therefore, the two operations of combined tillage and heavy harrow can only be completed by the tractors of 154.35 kW power, which caused the quantity of the tractors of 154.35 kW power to increase in a stepwise manner with the increase of the operation scale. The tractors of 66.15 kW power were suitable for production units of different operation scales, and their quantity increased in a stepwise manner with the increase of the operation scale. The total power per unit area of the tractor decreased with the increase of the operation scale, and it tended to stabilize when the operation scale exceeded 1600 hm$^2$.

In optimization by the MINP model, the combine harvesters of 239 kW power were not chosen for different operation scales. Since the small-power combine harvesters of

155 kW power had strong applicability to different operation scales, and the combine harvesters of 284 kW power had low cost per unit area, the numbers of the two power combine harvesters complemented each other. The total power per unit area of the combine harvesters decreased with the increase of the operation scale, and the total power per unit area tended to stabilize when the operation scale exceeded 1600 hm$^2$.

Considering the change in the total power per unit area of tractors and combine harvesters with the operation scale, the total power per unit area within the machinery systems decreased with the increase of the operation scale, and the total power per unit area tended to stabilize when the operation scale exceeded 1600 hm$^2$. The quantity of implements for different production scales are shown in Figure 5.

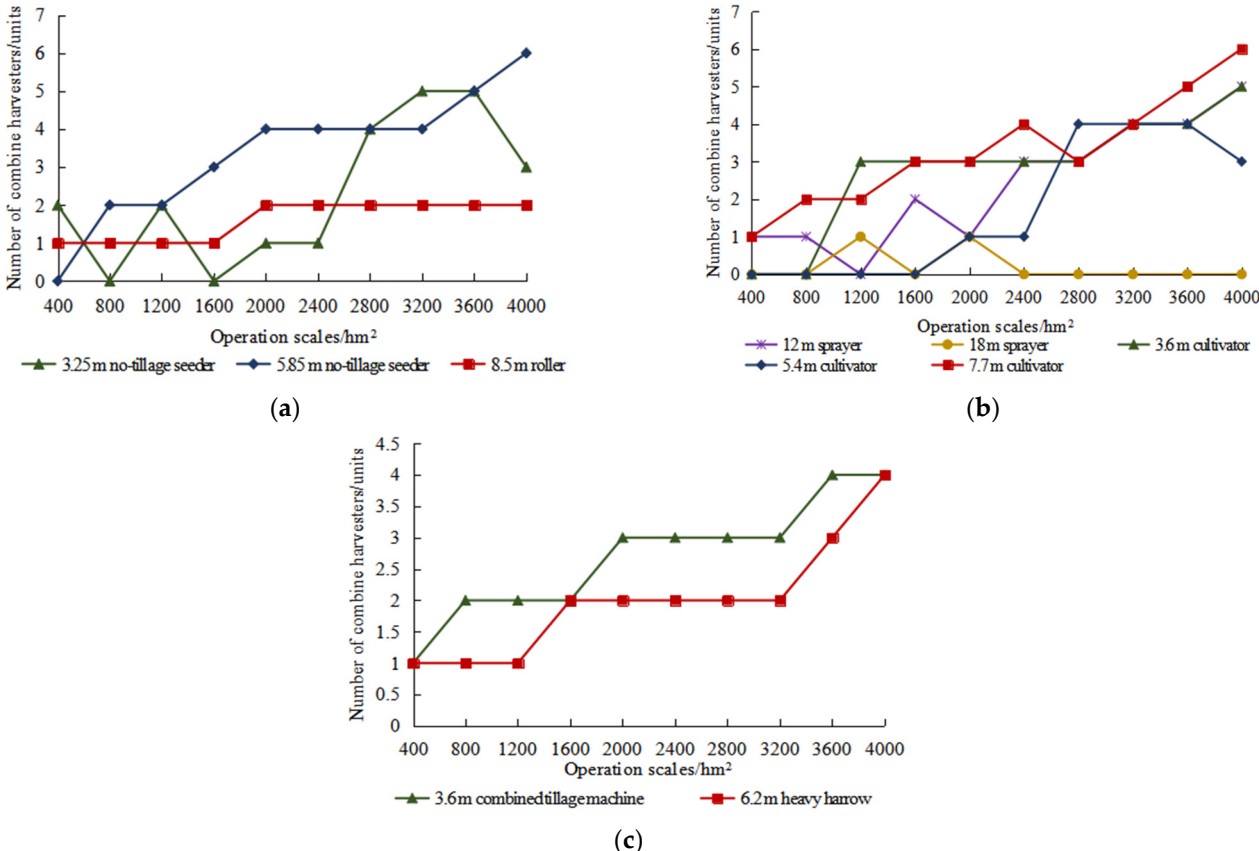

**Figure 5.** Number of implements. (**a**) Seeders and rollers; (**b**) Sprayers and cultivators; (**c**) Combined tillage machines and heavy harrows.

As could be seen from Figure 5, rolling and spraying had high operational efficiency, but sprayers participated in multiple operations, and the quantities of the two types of sprayers changed alternately with the increase in operation scale. The rollers were involved in fewer operations, and a relatively narrow roller of 8.5 m width could meet the operation requirements. Since the tractors of 271 kW power were not chosen in different production scales after optimization, only the combined tillage machines of 3.6 m width and heavy harrows of 6.2 m width, which form machinery units with the tractors of 154.35 kW power, were chosen among the two types of implements. Therefore, the quantities of the rollers of 8.5 m width, combined tillage machines of 3.6 m width, and heavy harrows of 6.2 m width increased in a stepwise manner with the increase in operation scale. In the machinery systems, two types of seeders and three types of cultivators appear in the optimization results of different operation scales. Among them, the seeders of 5.85 m width and cultivator of 7.7 m width only form operation machinery units with the tractors of 154.35 kW power, and the quantity change rule was the same as that of the tractors of 154.35 kW power, which

increased in a stepwise manner with the increase in operation scale. Operation costs per unit area for different operation scales are shown in Figure 6.

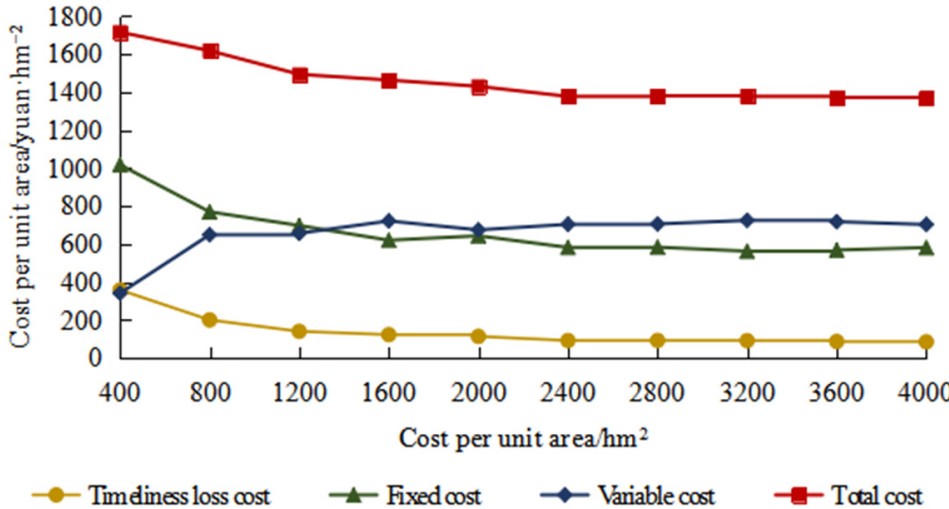

**Figure 6.** Operation costs for different operation scales.

As could be seen from Figure 6, as the operation scale increased, the per unit area timeliness loss cost and fixed cost gradually decreased, and the per unit area variable cost gradually increased. When the operation scale exceeded 1600 hm$^2$, the three costs tended to stabilize. Combining the three costs, the total cost per unit area gradually decreased with the increase in operation scale, and the total cost per unit area tended to stabilize when the operation scale exceeded 1600 hm$^2$.

As the operation scale gradient increased, the quantity of some types of machinery did not show a clear change rule, but the change in the total cost per unit area conformed to the actual rule, which verified the accuracy of the optimization results of the MINP model agricultural machinery systems. Production units could refer to the optimization results of agricultural machinery types and quantities for the different production scales as shown in Figures 4 and 5, and replace the agricultural machinery within the machinery systems to improve machinery utilization efficiency and economic benefits of the production units.

### 5. Conclusions

To address the current issues, a generalized MINP agricultural machinery system optimization model based on the timeliness losses of multiple operations was established. Timeliness loss rate functions for corn and soybean of sowing and harvesting within the research area were obtained through experiments. A 2000 hm$^2$ production unit employing a crop rotation and rotational tillage pattern, which was premised on no-tillage and sowing with straw mulching technology, was optimized applying the MINP model. Post-optimization improvements were observed in total power, total power per unit area, and total operation cost. The comparison of optimization results and their enhancement underpinned the necessity of model improvement. The MINP optimization model was applied to calculate the quantities of agricultural machinery for diverse operational scales, which provided plans for the allocation of agricultural machinery.

**Author Contributions:** Conceptualization, H.C.; methodology, J.Q. and J.S.; data curation, Y.Z. and J.S.; formal analysis, writing—original draft preparation, and visualization, J.S. and Y.Z.; Writing—review and editing, H.C., J.Q., J.S. and Y.Z.; funding acquisition, H.C. All authors have read and agreed to the published version of the manuscript.

**Funding:** Funding for this research was provided by the National Key Research and Development Program of China (Grant No. 2021YFD20004).

**Institutional Review Board Statement:** Not applicable.

**Informed Consent Statement:** Not applicable.

**Data Availability Statement:** Not applicable.

**Acknowledgments:** We are grateful for the time and effort provided by all farmers, hired workers, and other informants we interviewed in Heilongjiang Province of China. We thank all students and staff involved in the data collection for their hard work.

**Conflicts of Interest:** The authors declare no conflict of interest. The funders had no role in the design of the study; in the collection, analyses, or interpretation of data; in the writing of the manuscript, or in the decision to publish the results.

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
