# Peer review of "Optimization Model and Application for Agricultural Machinery Systems Based on Timeliness Losses of Multiple Operations"

_agriculture, doi:10.3390/agriculture13101969_

Round 1

Reviewer 1 Report

This paper presents an innovative enhancement of the nonlinear mixed-integer programming model for solving agricultural machinery system optimization problems. Experimental results are used to derive timeliness loss functions for typical crop projects, enhancing the alignment of optimization outcomes with real agricultural production. This also provides valuable references for selecting machine models and quantities for production units of varying operational scales. The article contains substantial content, and the reviewer recommends acceptance and publication.

However, there are certain issues that need to be addressed before considering it for publication.

1.      Line 23: In the subsequent text, please include specific Lingo software models for consistency throughout.

2.      Line 108-111: The sentences are overly long and complex, making them hard to comprehend. Consider breaking them into two shorter sentences.

3.      Line 117: There is a punctuation error in this line.

4.      Line 373: The timeliness loss rate of crops and their operational dates.

5.      Line 429-430: The model is solved using Lingo software. It is advisable to provide a brief overview of the software's key features at this point.

6.      Line 538: The sequence of sowing and post-sow rolling operations in these plots has been reversed.

7.      Table 1: The font size in the table text is inconsistent and needs adjustment.

8.      Table 2: The table illustrates changes in the number of machines after optimization. It is advisable to label the cost in a similar manner for better reader comprehension.

If the issues mentioned are adequately addressed, the reviewer believes that this paper's essential contributions are significant in addressing the optimization problems in agricultural machinery systems.

Most of the English writing is well written and only individual words need to be revised.

Reviewer 2 Report

1. The abstract needs to be further refined, such as the proposed MINP model, what are the key problems to be solved? How was it resolved?

2. Corn and soybeans as examples, or does the model only apply to those two crops? Are the critical operations for corn and soybeans of the whole growing season? These problems should be stated in the Abstract.

3. According to the introduction, the current methods for optimizing agricultural machinery systems are mainly nonlinear programming, with less linear programming and little relevance to this paper. It is suggested to compress the content of paragraphs 2 and 3 and focus on nonlinear programming in Introduction section.

4. P67 What does the reference of Zhou Yingchao mean? Is the loss rate coefficients better than timeliness?

5. Line 108 What does the constraints refer to

6. What’s the difficulty for regulations of timeliness loss for multiple key crop operations? And how does the paper plan to solve it? These questions should be introduced in the Introduction section.

7. Mixed Integer Nonlinear Programming is the core of the model, why to select this model? What’s the advantage of this method? The method should be introduced in the Introduction.

8. The text from line 119-125 should be introduced in Introduction section.

Reviewer 3 Report

  The paper shows improvements of agricultural machinery system optimization model method based on the nonlinear programming model which have theory and practice significance.

Firstly, timeless loss cost mathematic model was established and added to the objective function. Secondly, sequence constraint functions of key operations date were designed and added in constrains of the model. These are obviously innovating improvements for the kind of model in the field which are the existed problems perplexing researchers for years. On the other hand, experiments were also designed ang carried out to obtain practice timeless loss functions of four operations of two crops which powerfully support the paper to achieve more efficient schemes for the concrete production unit. On the basis of which, optimized schemes various with production scales were also put forward and discussed which make the content of the paper more completed and more widely . The paper is worth to be hired and published.   There also some weaknesses in the paper, which are waiting for being corrected or refined. Some of them are as follows:

Line 2-4: The title cannot accurately express the contents of the paper. It seems better if changing it to “Optimization Model and Application for Agricultural Machinery Systems Based on Mixed Integer Nonlinear Programming Considering Timeliness Losses of Multiple Operations”.

Line 17-21 in Abstract: It is better to use the passive voice in the past tense to express research contents in the paper. So the sentence should be corrected following the rule. There also exist same misexpressions in the other parts of the paper. It is suggested that authors can find them and correct.

Line 105-116: The key innovations of the paper should be clearly expressed in this paragraph after analyzing literatures.

Line 122-126: It should be clearly stated that the model will become a nonlinear programming model after adding a timeliness loss function.

Line 133: It is suggested that authors should analyze exact meanings of some concepts or terms, like machine, machinery, machine group etal. which are the foundations for the clearly writing and expressing.

Line 166: “residual value” should to be replaced by “salvage value”. Other locations where this issue occurs also need to be modified

Line 170-171: It is better that the term “original value” being replaced with “purchasing price”, and “residual value” being replaced with “ salvage value”.

Line 288-289: It is better to change the title to “Agricultural Machinery System Optimization Based on MINP model for Large-Scale production unit”.

Line 362: It is better to change the title to “Determination of Timeless Loss Function of Key Operations”. It is also necessary to add the necessity information to Figure 1. making it more accurate.

Table 1 following line 412: Pay attention to the consistency of font and size in the table. Express dimensions of each parameter correctly. Check and correct table 2 below, if there is similar misexpressions.

Line 417: The word “include” appeared twice in the sentence and sentence logic is confusion. Suggesting to correct it.

Line 467-470, Line 498-504: The sentences are too long and difficult to understand. Please break it down into short sentences and be sure the meaning of new sentences being expressed clearly.

Line 567-576: Names and parameters of the implements don’t meet the standard expressions. Suggesting to correct.  

Some concepts and sentences are not very clear to expresss the content of the paper, It is suggested for author to correct according the comments in the above column.

Reviewer 4 Report

1. Similar work has been done by authors in the Ref. [29], including in four-dimensional subscript variable setting, objective function and constraints. This manuscript should further shed new insights on research opinion and method. So what are the improved or enhanced differences between the manuscript and the Ref. [29]?

2. In Line 200, the title of section 2.2.3 is the same as that of section 2.2.2.

3. In Line 215, the timeliness loss rate function ylp(t) is present firstly. According to In Line 362key operation timeliness loss function is obtained by a nonlinear regression analysis. What are the key factors influencing the timeliness loss rate and how the key factor affect the timeliness loss?

4. In Line 221, equation (6) is the same as equation (5) where variable Sq does not exist. Then operation area constraint model should be revised.

5. In Line 408, Figure 3 is an error.

6. In Line 412, several variables have error units in Table 1.

7. In Line 119, ‘Owing to the differences in crop varieties, natural environments, and other factors, each crop has its unique growth and maturity cycle.’ It can be seen that the timeliness losses are relative to the optimal date for crop sowing and harvesting, and the optimal date is indeterminate. So how to decide the optimal date for agricultural operations, which is mentioned in equation (12).

8. In this manuscript, the optimization model of agricultural machinery systems based on timeliness loss of multiple operations is established. It is crucial to solve the optimization model. In Line 421, the Ref. [35] cannot elucidate the calculating process which is worthy of explanation.

There are several errors in the manuscripts.

Round 2

Reviewer 2 Report

Revision is satisfied.

Author Response

感谢您的认可!

Reviewer 4 Report

1. In order to understand the optimization model, it is necessary to analyze and discuss the key factors influencing the times loss of crops.

2. If the calculating process of MINP cannot be discussed and explained, it is suggested that MINP is deleted in the title of the manuscript.

Minor improvement.

Author Response

请参阅附件。
